# Improved memory in recurrent neural networks with sequential non-normal dynamics

**A. Emin Orhan[1] and Xaq Pitkow[2,3]**
[1]New York University (eo41@nyu.edu), [2]Rice University, [3]Baylor College of Medicine

## Abstract

Training recurrent neural networks (RNNs) is a hard problem due to degeneracies in the optimization landscape, a problem also known as vanishing/exploding gradients. Short of designing new RNN architectures, previous methods for dealing with this problem usually boil down to orthogonalization of the recurrent dynamics, either at initialization or during the entire training period. The basic motivation behind these methods is that orthogonal transformations are isometries of the Euclidean space, hence they preserve (Euclidean) norms and effectively deal with vanishing/exploding gradients. However, this ignores the crucial effects of *non-linearity* and *noise*. In the presence of a non-linearity, orthogonal transformations no longer preserve norms, suggesting that alternative transformations might be better suited to non-linear networks. Moreover, in the presence of noise, norm preservation itself ceases to be the ideal objective. A more sensible objective is maximizing the signal-to-noise ratio (SNR) of the propagated signal instead. Previous work has shown that in the linear case, recurrent networks that maximize the SNR display strongly non-normal, sequential dynamics and orthogonal networks are highly suboptimal by this measure. Motivated by this finding, here we investigate the potential of non-normal RNNs, i.e. RNNs with a non-normal recurrent connectivity matrix, in sequential processing tasks. Our experimental results show that non-normal RNNs outperform their orthogonal counterparts in a diverse range of benchmarks. We also find evidence for increased non-normality and hidden chain-like feedforward motifs in trained RNNs initialized with orthogonal recurrent connectivity matrices.

## 1 Introduction

Modeling long-term dependencies with recurrent neural networks (RNNs) is a hard problem due to degeneracies inherent in the optimization landscapes of these models, a problem also known as the vanishing/exploding gradients problem (Hochreiter, 1991; Bengio et al., 1994). One approach to addressing this problem has been designing new RNN architectures that are less prone to such difficulties, hence are better able to capture long-term dependencies in sequential data (Hochreiter & Schmidhuber, 1997; Cho et al., 2014; Chang et al., 2017; Bai et al., 2018). An alternative approach is to stick with the basic vanilla RNN architecture instead, but to constrain its dynamics in some way so as to eliminate or reduce the degeneracies that otherwise afflict the optimization landscape. Previous proposals belonging to this second category generally boil down to orthogonalization of the recurrent dynamics, either at initialization or during the entire training period (Le et al., 2015; Arjovsky et al., 2016; Wisdom et al., 2016). The basic idea behind these methods is that orthogonal transformations are isometries of the Euclidean space, hence they preserve distances and norms, which enables them to deal effectively with the vanishing/exploding gradients problem.

However, this idea ignores the crucial effects of *non-linearity* and *noise*. Orthogonal transformations no longer preserve distances and norms in the presence of a non-linearity, suggesting that alternative transformations might be better suited to non-linear networks (this point was noted by Pennington et al. (2017) and Chen et al. (2018) before, where isometric initializations that take the non-linearity into account were proposed). Similarly, in the presence of noise, norm preservation itself ceases to be the ideal objective. One must instead maximize the signal-to-noise ratio (SNR) of the propagated signal. In neural networks, noise comes in both through the stochasticity of the stochastic gradient descent (SGD) algorithm and sometimes also through direct noise injection for regularization purposes, as

in dropout (Srivastava et al., 2014). Previous work has shown that even in a simple linear setting, recurrent networks that maximize the SNR display strongly non-normal, sequential dynamics and orthogonal networks are highly suboptimal by this measure (Ganguli et al., 2008).

Motivated by these observations, in this paper, we investigate the potential of non-normal RNNs, i.e. RNNs with a non-normal recurrent connectivity matrix, in sequential processing tasks. Recall that a normal matrix is a matrix with an orthonormal set of eigenvectors, whereas a non-normal matrix does not have an orthonormal set of eigenvectors. This property allows non-normal systems to display interesting transient behaviors that are not available in normal systems. This kind of transient behavior, specifically a particular kind of transient amplification of the signal in certain non-normal systems, underlies their superior memory properties (Ganguli et al., 2008), as will be discussed further below. Our empirical results show that non-normal vanilla RNNs significantly outperform their orthogonal counterparts in a diverse range of benchmarks.[1]

## 2 BACKGROUND

### 2.1 MEMORY IN LINEAR RECURRENT NETWORKS WITH NOISE

Ganguli et al. (2008) studied memory properties of linear recurrent networks injected with a scalar temporal signal $s_t$, and noise $\mathbf{z}_t$:

$$\mathbf{h}_t = \mathbf{W}\mathbf{h}_{t-1} + \mathbf{v}s_t + \mathbf{z}_t \tag{1}$$

The noise is assumed to be *i.i.d.* with $\mathbf{z}_t \sim \mathcal{N}(0, \mathbf{I})$. Ganguli et al. (2008) then analyzed the Fisher memory matrix (FMM) of this system, defined as:

$$\mathbf{J}_{kl}(s_{\leq t}) = \left\langle -\frac{\partial^2}{\partial s_{t-k} \partial s_{t-l}} \log p(\mathbf{h}_t|s_{\leq t}) \right\rangle_{p(\mathbf{h}_t|s_{\leq t})} \tag{2}$$

For linear networks with Gaussian noise, it is easy to show that $\mathbf{J}_{kl}(s_{\leq t})$ is, in fact, independent of the past signal history $s_{\leq t}$. Ganguli et al. (2008) specifically analyzed the diagonal of the FMM: $J(k) \equiv \mathbf{J}_{kk}$, which can be written explicitly as:

$$J(k) = \mathbf{v}^\top \mathbf{W}^{k\top} \mathbf{C}^{-1} \mathbf{W}^k \mathbf{v} \tag{3}$$

where $\mathbf{C} = \sum_{k=0}^{\infty} \mathbf{W}^k \mathbf{W}^{k\top}$ is the noise covariance matrix, and the norm of $\mathbf{W}^k \mathbf{v}$ can be roughly thought of as representing the signal strength. The total Fisher memory is the sum of $J(k)$ over all past time steps $k$:

$$J_{\text{tot}} = \sum_{k=0}^{\infty} J(k) \tag{4}$$

Intuitively, $J(k)$ measures the information contained in the current state of the system, $\mathbf{h}_t$, about a signal that entered the system $k$ time steps ago, $s_{t-k}$. $J_{\text{tot}}$ is then a measure of the total information contained in the current state of the system about the entire past signal history, $s_{\leq t}$.

The main result in Ganguli et al. (2008) shows that $J_{\text{tot}} = 1$ for *all* normal matrices $\mathbf{W}$ (including all orthogonal matrices), whereas in general $J_{\text{tot}} \leq N$, where $N$ is the network size. Remarkably, the memory upper bound can be achieved by certain highly non-normal systems and several examples are explicitly given in Ganguli et al. (2008). Two of those examples are illustrated in Figure 1a (right): a uni-directional "chain" network and a chain network with feedback. In the chain network, the recurrent connectivity is given by $\mathbf{W}_{ij} = \alpha \delta_{j,i-1}$ and in the chain with feedback network, it is given by $\mathbf{W}_{ij} = \alpha \delta_{j,i-1} + \beta \delta_{j,i+1}$, where $\alpha$ and $\beta$ are the feedforward and feedback connection weights, respectively (here $\delta$ denotes the Kronecker delta function). In addition, in order to achieve optimal memory, the signal must be fed at the source neuron in these networks, i.e. $\mathbf{v} = [1, 0, 0, \dots, 0]^\top$.

Figure 1b compares the Fisher memory curves, $J(k)$, of these non-normal networks with the Fisher memory curves of two example normal networks, namely recurrent networks with identity or random orthogonal connectivity matrices. The two non-normal networks have extensive memory capacity, i.e. $J_{\text{tot}} \sim O(N)$, whereas for the normal examples, $J_{\text{tot}} = 1$. The crucial property that enables extensive memory in non-normal networks is *transient amplification*: after the signal enters the network, it is amplified supralinearly for a time of length $O(N)$ before it eventually dies out (Figure 1c). This kind of transient amplification is not possible in normal networks.

---

[1]Code available at: `https://github.com/eminorhan/nonnormal-init`

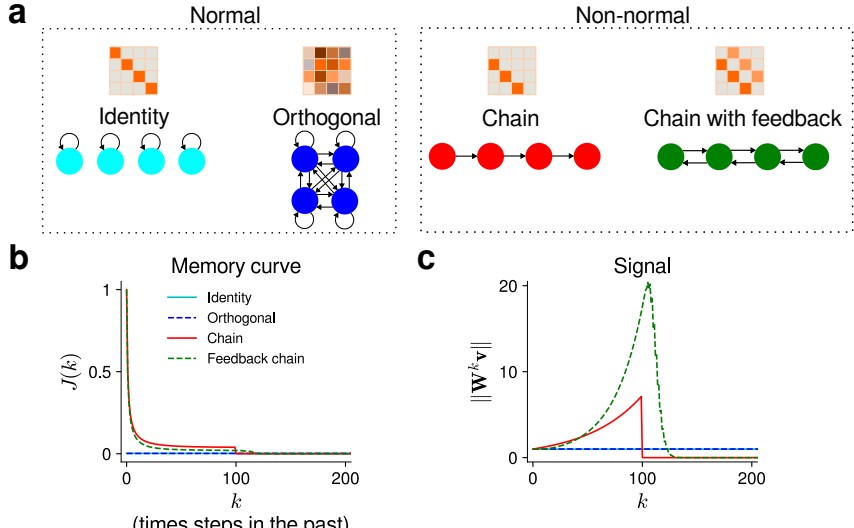

Figure 1: **a** Schematic diagrams of different recurrent networks and the corresponding recurrent connectivity matrices (upper panel). **b** Memory curves, $J(k)$ (Equation 3), for the four recurrent networks shown in **a**. The non-normal networks, chain and chain with feedback, have extensive memory capacity: $J_{\text{tot}} \sim O(N)$, whereas the normal networks, identity and random orthogonal, have $J_{\text{tot}} = 1$. **c** Extensive memory is made possible in non-normal networks by *transient amplification*: the signal is amplified for a time of length $O(N)$ before it dies out, abruptly in the case of the chain network and more gradually in the case of the chain network with feedback. In **b** and **c**, the network size is $N = 100$ for all four networks.

## 2.2 A TOY NON-LINEAR EXAMPLE: NON-LINEARITY AND NOISE INDUCE SIMILAR EFFECTS

The preceding analysis by Ganguli et al. (2008) is exact in linear networks. Analysis becomes more difficult in the presence of a non-linearity. However, we now demonstrate that the non-normal networks shown in Figure 1a have advantages that extend beyond the linear case. The advantages in the non-linear case are due to reduced interference in these non-normal networks between signals entering the network at different time points in the past.

To demonstrate this with a simple example, we will ignore the effect of noise for now and consider the effect of non-linearity on the linear decodability of past signals from the current network activity. We thus consider deterministic non-linear networks of the form (see Appendix A for additional details):

$$\mathbf{h}_t = f(\mathbf{W}\mathbf{h}_{t-1} + \mathbf{v}s_t) \tag{5}$$

and ask how well we can linearly decode a signal that entered the network $k$ time steps ago, $s_{t-k}$, from the current activity of the network, $\mathbf{h}_t$. Figure 2c compares the decoding performance in a non-linear orthogonal network with the decoding performance in the non-linear chain network. Just as in the linear case with noise (Figure 2b), the chain network outperforms the orthogonal network.

To understand intuitively why this is the case, consider a chain network with $\mathbf{W}_{ij} = \delta_{j,i-1}$ and $\mathbf{v} = [1, 0, 0, \ldots, 0]^\top$. In this model, the responses of the $N$ neurons after $N$ time steps (at $t = N$) are given by $f(s_N), f(f(s_{N-1})), ..., f(f(\ldots f(s_1)\ldots))$, respectively, starting from the source neuron. Although the non-linearity $f(\cdot)$ makes perfect linear decoding of the past signal $s_{t-k}$ impossible, one may still imagine being able to decode the past signal with reasonable accuracy as long as $f(\cdot)$ is not "too non-linear". A similar intuition holds for the chain network with feedback as well, as long as the feedforward connection weight, $\alpha$, is sufficiently stronger than the feedback connection strength, $\beta$. A condition like this must already be satisfied if the network is to maintain its optimal memory properties and also be dynamically stable at the same time (Ganguli et al., 2008).

In normal networks, however, linear decoding is further degraded by interference from signals entering the network at different time points, in addition to the degradation caused by the non-linearity. This is easiest to see in the identity network (a similar argument holds for the random orthogonal example too), where the responses of the neurons after $N$ time steps are identically given

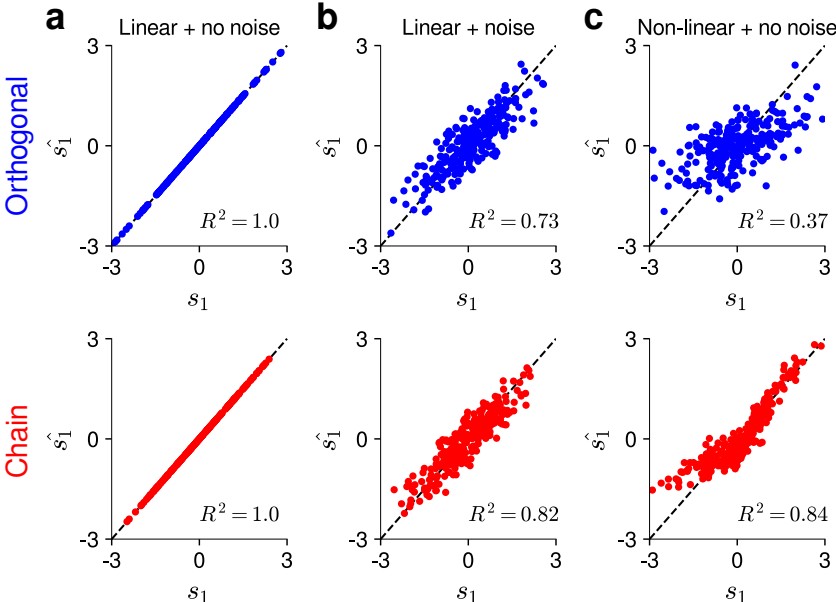

Figure 2: Linear decoding experiments. **a** In a linear network with no noise, the past signal $s_1$ can be perfectly reconstructed from the current activity vector $\mathbf{h}_{100}$ using a linear decoder. **b** When noise is added, the chain network outperforms the orthogonal network as predicted from the theory in Ganguli et al. (2008). **c** In a completely deterministic system, introducing a non-linearity has a similar effect to that of noise. The chain network again outperforms the orthogonal one when the signal is reconstructed with a linear decoder. As discussed further in the text, this is because the signal is subject to more interference in the orthogonal network than in the chain network. All simulations in this figure used networks with $N = 100$ recurrent units. In **c**, we used the `elu` non-linearity for $f(\cdot)$ (Clevert et al., 2016). For the chain network, we assume that the signal is fed at the source neuron.

by $f(f(\ldots f(f(s_1) + s_2) \ldots) + s_N)$, if one assumes $\mathbf{v} = [1, 1, 1, \ldots, 1]^\top$. Linear decoding is harder in this case, because a signal $s_{t-k}$ is both distorted by multiple steps of non-linearity and also mixed with signals entering at other time points.

## 3 RESULTS

### 3.1 EXPERIMENTS

Because assuming an *a priori* fixed non-normal structure for an RNN runs the risk of being too restrictive, in this paper, we instead explore the promise of non-normal networks as *initializers* for RNNs. Throughout the paper, we will be primarily comparing the four RNN architectures schematically depicted in Figure 1a as initializers: two of them normal networks (identity and random orthogonal) and the other two non-normal networks (chain and chain with feedback), the last two being motivated by their optimal memory properties in the linear case, as reviewed above.

#### 3.1.1 COPY, ADDITION, PERMUTED SEQUENTIAL MNIST

Copy, addition, and permuted sequential MNIST tasks were commonly used as benchmarks in previous RNN studies (Arjovsky et al., 2016; Bai et al., 2018; Chang et al., 2017; Hochreiter & Schmidhuber, 1997; Le et al., 2015; Wisdom et al., 2016). We now briefly describe each of these tasks.

**Copy task:** The input is a sequence of integers of length $T$. The first 10 integers in the sequence define the target subsequence that is to be copied and consist of integers between 1 and 8 (inclusive). The next $T - 21$ integers are set to 0. The integer after that is set to 9, which acts as the cue indicating that the model should start copying the target subsequence. The final 10 integers are set to 0. The

output sequence that the model is trained to reproduce consists of $T - 10$ 0s followed by the target subsequence from the input that is to be copied. To make sure that the task requires a sufficiently long memory capacity, we used a large sequence length, $T = 500$, comparable to the largest sequence length considered in Arjovsky et al. (2016) for the same task.

**Addition task:** The input consists of two sequences of length $T$. The first one is a sequence of random numbers drawn uniformly from the interval $[0, 1]$. The second sequence is an indicator sequence with 1s at exactly two positions and 0s everywhere else. The positions of the two 1s indicate the positions of the numbers to be added in the first sequence. The target output is the sum of the two corresponding numbers. The position of the first 1 is drawn uniformly from the first half of the sequence and the position of the second 1 is drawn uniformly from the second half of the sequence. Again, to ensure that the task requires a sufficiently long memory capacity, we chose $T = 750$, which is the same as the largest sequence length considered in Arjovsky et al. (2016) for the same task.

**Permuted sequential MNIST (psMNIST):** This is a sequential version of the standard MNIST benchmark where the pixels are fed to the model one pixel at a time. To make the task hard enough, we used the permuted version of the sequential MNIST task where a fixed random permutation is applied to the pixels to eliminate any spatial structure before they are fed into the model.

We used vanilla RNNs with $N = 25$ recurrent units in the psMNIST task and $N = 100$ recurrent units in the copy and addition tasks. We used the `elu` nonlinearity for the copy and the psMNIST tasks (Clevert et al., 2016), and the `relu` nonlinearity for the addition problem (because `relu` proved to be more natural for remembering positive numbers). Batch size was 16 in all tasks.

As mentioned above, the scaled identity and the scaled random orthogonal networks constituted the normal initializers. In the scaled identity initializer, the recurrent connectivity matrix was initialized as $\mathbf{W} = \lambda \mathbf{I}$ and the input matrix $\mathbf{V}$ was initialized as $\mathbf{V}_{ij} \sim \mathcal{N}(0, 0.9/\sqrt{N})$. In the random orthogonal initializer, the recurrent connectivity matrix was initialized as $\mathbf{W} = \lambda \mathbf{Q}$, where $\mathbf{Q}$ is a random dense orthogonal matrix, and the input matrix $\mathbf{V}$ was initialized in the same way as in the identity initializer.

The feedforward chain and the chain with feedback networks constituted our non-normal initializers. In the chain initializer, the recurrent connectivity matrix was initialized as $\mathbf{W}_{ij} = \alpha \delta_{j,i-1}$ and the input matrix $\mathbf{V}$ was initialized as $\mathbf{V} \sim 0.9 \mathbf{I}_{N \times d}$, where $\mathbf{I}_{N \times d}$ denotes the $N \times d$-dimensional identity matrix. Note that this choice of $\mathbf{V}$ is a natural generalization of the the source injecting input vector that was found to be optimal in the linear case with scalar signals to multi-dimensional inputs (as long as $N \gg d$). In the chain with feedback initializer, the recurrent connectivity matrix was initialized as $\mathbf{W}_{ij} = 0.99 \delta_{j,i-1} + \beta \delta_{j,i+1}$ and the input matrix $\mathbf{V}$ was initialized in the same way as in the chain initializer.

We used the rmsprop optimizer for all models, which we found to be the best method for this set of tasks. The learning rate of the optimizer was a hyperparameter which we tuned separately for each model and each task. The following learning rates were considered in the hyper-parameter search: $8 \times 10^{-4}, 5 \times 10^{-4}, 3 \times 10^{-4}, 10^{-4}, 8 \times 10^{-5}, 5 \times 10^{-5}, 3 \times 10^{-5}, 10^{-5}, 8 \times 10^{-6}, 5 \times 10^{-6}, 3 \times 10^{-6}$. We ran each model on each task 6 times using the integers from 1 to 6 as random seeds.

In addition, the following model-specific hyperparameters were searched over for each task:

Chain: feedforward connection weight, $\alpha \in \{0.99, 1.00, 1.01, 1.02, 1.03, 1.04, 1.05\}$.

Chain with feedback: feedback connection weight, $\beta \in \{0.01, 0.02, 0.03, 0.04, 0.05, 0.06, 0.07\}$.

Scaled identity: scale, $\lambda \in \{0.01, 0.96, 0.99, 1.0, 1.01, 1.02, 1.03, 1.04, 1.05\}$.

Random orthogonal: scale, $\lambda \in \{0.01, 0.96, 0.99, 1.0, 1.01, 1.02, 1.03, 1.04, 1.05\}$.

This yields a total of $7 \times 11 \times 6 = 462$ different runs for each experiment in the non-normal models and a total of $9 \times 11 \times 6 = 594$ different runs in the normal models. Note that we ran more extensive hyper-parameter searches for the normal models than for the non-normal models in this set of tasks.

Figure 3a-c shows the validation losses for each model with the best hyper-parameter settings. The non-normal initializers generally outperform the normal initializers. Figure 3d-f shows for each model the number of "successful" runs that converged to a validation loss below a criterion level (which we set to be 50% of the loss for a baseline random model). The chain model outperformed all other models by this measure (despite having a smaller total number of runs than the normal models).

In the copy task, for example, none of the runs for the normal models was able to achieve the criterion level, whereas 46 out of 462 runs for the chain model and 11 out of 462 runs for the feedback chain model reached the criterion loss (see Appendices B & C for further results and discussion).

### 3.1.2 LANGUAGE MODELING EXPERIMENTS

To investigate if the benefits of non-normal initializers extend to more realistic problems, we conducted experiments with three standard language modeling tasks: word-level Penn Treebank (PTB), character-level PTB, and character-level `enwik8` benchmarks.

For the language modeling experiments in this subsection, we used the code base provided by Salesforce Research (Merity et al., 2018a;b): `https://github.com/salesforce/awd-lstm-lm`. We refer the reader to Merity et al. (2018a;b) for a more detailed description of the benchmarks. For the experiments in this subsection, we generally preserved the model setup used in Merity et al. (2018a;b), except for the following differences: 1) We replaced the gated RNN architectures (LSTMs and QRNNs) used in Merity et al. (2018a;b) with vanilla RNNs; 2) We observed that vanilla RNNs require weaker regularization than gated RNN architectures. Therefore, in the word-level PTB task, we set all dropout rates to $0.1$. In the character-level PTB task, all dropout rates except `dropoute` were set to $0.1$, which was set to $0$. In the `enwik8` benchmark, all dropout rates were set to $0$; 3) We trained the word-level PTB models for 60 epochs, the character-level PTB models for 500 epochs and the `enwik8` models for 35 epochs.

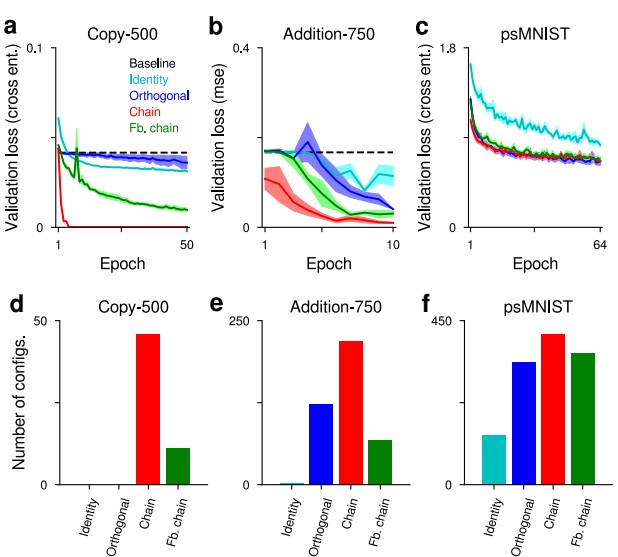

Figure 3: Results on copy, addition, and psMNIST benchmarks. **a-c** Validation losses with the best hyper-parameter settings. Solid lines are the means and shaded regions are standard errors over different runs using different random seeeds. For the copy and addition tasks, we also show the loss values for random baseline models (dashed lines). For the psMNIST task, the mean cross-entropy loss for a random classifier is $\log(10) \approx 2.3$, thus all four models comfortably outperform this random baseline right from the end of the first training epoch. **d-f** Number of "successful" runs (or hyperparameter configurations) that converged to a validation loss below 50% of the loss for the random baseline model. Note that the total number of runs was higher for the normal models vs. the non-normal models (594 vs. 462 runs per experiment). Despite this, the non-normal models generally outperformed the normal models even by this measure.

We compared the same four models described in the previous subsection. As in Merity et al. (2018a), we used the Adam optimizer and thus only optimized the $\alpha$, $\beta$, $\lambda$ hyper-parameters for the experiments in this subsection. For the hyper-parameter $\alpha$ in the chain model and the hyper-parameter $\lambda$ in the scaled identity and random orthogonal models, we searched over 21 values uniformly spaced between $0.05$ and $1.05$ (inclusive); whereas for the chain with feedback model, we set the feedforward connection weight, $\alpha$, to the optimal value it had in the chain model and searched over 21 $\beta$ values uniformly spaced between $0.01$ and $0.21$ (inclusive). In addition, we repeated each experiment 3 times using different random seeds, yielding a total of 63 runs for each model and each benchmark.

The results are shown in Figure 4 and in Table 1. Figure 4 shows the validation loss over the course of training in units of bits per character (bpc). Table 1 reports the test losses at the end of training. The non-normal models outperform the normal models on the word-level and character-level PTB benchmarks. The differences between the models are less clear on the `enwik8` benchmark. However, in terms of the test loss, the non-normal feedback chain model outperforms the other models on all three benchmarks (Table 1).

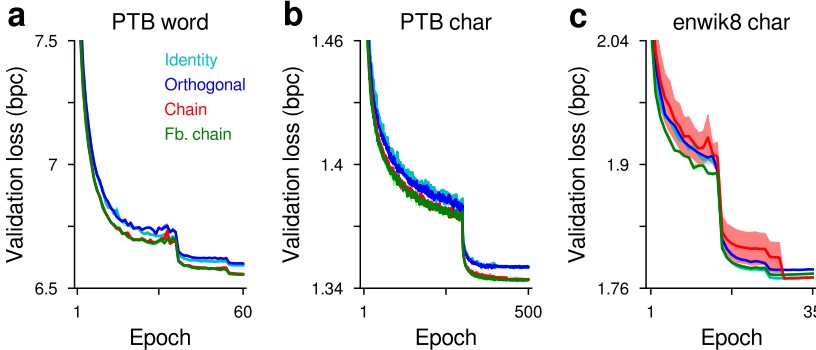

Figure 4: Results on language modeling benchmarks. Solid lines are the means and shaded regions are standard errors over 3 different runs using different random seeeds.

Table 1: Test losses (bpc) on language modeling benchmarks. The numbers represent mean ± s.e.m. over 3 independent runs. LSTM results are from Merity et al. (2018a;b).

| MODEL | PTB WORD | PTB CHAR. | ENWIK8 |
|---|---|---|---|
| IDENTITY | $6.550 \pm 0.002$ | $1.312 \pm 0.000$ | $1.783 \pm 0.003$ |
| ORTHO. | $6.557 \pm 0.002$ | $1.312 \pm 0.001$ | $1.843 \pm 0.046$ |
| CHAIN | $6.514 \pm 0.001$ | $1.308 \pm 0.000$ | $1.803 \pm 0.017$ |
| FB. CHAIN | $\mathbf{6.510 \pm 0.001}$ | $\mathbf{1.307 \pm 0.000}$ | $\mathbf{1.774 \pm 0.002}$ |
| 3-LAYER LSTM | 5.878 | 1.175 | 1.232 |

We note that the vanilla RNN models perform significantly worse than the gated RNN architectures considered in Merity et al. (2018a;b). We conjecture that this is because gated architectures are generally better at modeling contextual dependencies, hence they have inductive biases better suited to language modeling tasks. The primary benefit of non-normal dynamics, on the other hand, is enabling a longer memory capacity. Below, we will discuss whether non-normal dynamics can be used in gated RNN architectures to improve performance as well.

## 3.2 HIDDEN FEEDFORWARD STRUCTURES IN TRAINED RNNS

We observed that training made vanilla RNNs initialized with orthogonal recurrent connectivity matrices non-normal. We quantified the non-normality of the trained recurrent connectivity matrices using a measure introduced by Henrici (1962): $d(\mathbf{W}) \equiv \sqrt{\|\mathbf{W}\|_{\mathrm{F}}^2 - \sum_i |\lambda_i|^2}$, where $\| \cdot \|_{\mathrm{F}}$ denotes the Frobenius norm and $\lambda_i$ is the $i$-th eigenvalue of $\mathbf{W}$. This measure equals 0 for all normal matrices and is positive for non-normal matrices. We found that $d(\mathbf{W})$ became positive for all successfully trained RNNs initialized with orthogonal recurrent connectivity matrices. Table 2 reports the aggregate statistics of $d(\mathbf{W})$ for orthogonally initialized RNNs trained on the toy benchmarks.

Although increased non-normality in trained RNNs is an interesting observation, the Henrici index, by itself, does not tell us what structural features in trained RNNs contribute to this increased non-normality. Given the benefits of chain-like feedforward non-normal structures in RNNs for improved memory, we hypothesized that training might have installed hidden chain-like feedforward structures in trained RNNs and that these feedforward structures were responsible for their increased non-normality.

To uncover these hidden feedforward structures, we performed an analysis suggested by Rajan et al. (2016). In this analysis, we first injected a unit pulse of input to the network at the beginning of the trial and let the network evolve for 100 time steps afterwards according to its recurrent dynamics with no direct input. We then ordered the recurrent units by the time of their peak activity (using a small amount of jitter to break potential ties between units) and plotted the mean recurrent connection

Table 2: Henrici indices, $d(\mathbf{W})$, of trained RNNs initialized with orthogonal recurrent connectivity matrices. The numbers represent mean $\pm$ s.e.m. over all successfully trained networks. We define training success as having a validation loss below 50% of a random baseline model. Note that by this measure, none of the orthogonally initialized RNNs was successful on the copy task (Figure 3d).

| TASK | IDENTITY | ORTHOGONAL |
|---|---|---|
| ADDITION-750 | $2.33 \pm 1.02$ | $2.74 \pm 0.07$ |
| PSMNIST | $1.01 \pm 0.12$ | $2.72 \pm 0.08$ |

weights, $\mathbf{W}_{ij}$, as a function of the order difference between two units, $i - j$. Positive $i - j$ values correspond to connections from earlier peaking units to later peaking units, and vice versa for negative $i - j$ values. In trained RNNs, the mean recurrent weight profile as a function of $i - j$ had an asymmetric peak, with connections in the "forward" direction being, on average, stronger than those in the opposite direction. Figure 5 shows examples with orthogonally initialized RNNs trained on the addition and the permuted sequential MNIST tasks. Note that for a purely feedforward chain, the weight profile would have a single peak at $i - j = 1$ and would be zero elsewhere. Although the weight profiles for trained RNNs are not this extreme, the prominent asymmetric bump with a peak at a positive $i - j$ value indicates a hidden chain-like feedforward structure in these networks.

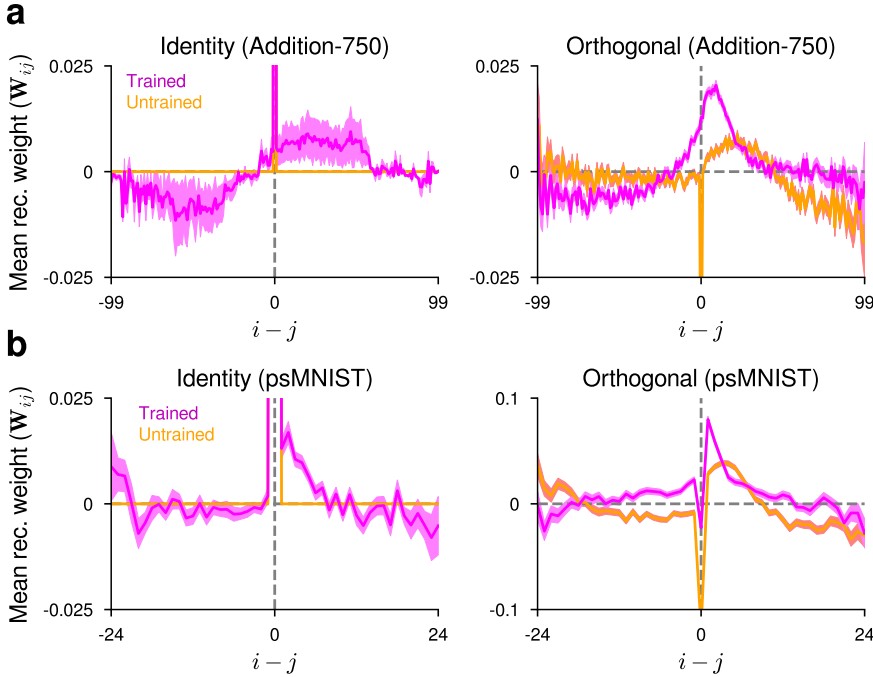

Figure 5: Training induces hidden chain-like feedforward structures in vanilla RNNs. The units are first ordered by the time of their peak activity. Then, the mean recurrent connection weight is plotted as a function of the order difference between two units, $i - j$. Results are shown for RNNs trained on the addition (**a**) and the permuted sequential MNIST (**b**) tasks. The left column shows the results for RNNs initialized with a scaled identity matrix, the right column shows the results for RNNs initialized with random orthogonal matrices. In each case, training induces hidden chain-like feedforward structures in the networks, as indicated by an asymmetric bump peaked at a positive $i - j$ value in the weight profile. This kind of structure is either non-existent (identity) or much less prominent (orthogonal) in the initial untrained networks. For the results shown here, we only considered sufficiently well-trained networks that achieved a validation loss below 50% of the loss for a baseline random model at the end of training. The solid lines and shaded regions represent means and standard errors of the mean weight profiles over these networks.

Table 3: Test losses (bpc) on language modeling benchmarks using 3-layer LSTMs (adapted from Merity et al. (2018a;b)) with different initialization schemes. Other experimental details were identical to those described in 3.1.2 above. The numbers represent mean $\pm$ s.e.m. over 3 independent runs.

| MODEL | PTB WORD | PTB CHAR. | ENWIK8 |
|---|---|---|---|
| ORTHO. | $5.937 \pm 0.002$ | $1.230 \pm 0.001$ | $1.583 \pm 0.001$ |
| CHAIN | $\mathbf{5.935 \pm 0.001}$ | $1.230 \pm 0.001$ | $1.586 \pm 0.000$ |
| PLAIN | $5.949 \pm 0.007$ | $1.245 \pm 0.001$ | $1.584 \pm 0.002$ |
| MIXED | $5.944 \pm 0.004$ | $\mathbf{1.227 \pm 0.000}$ | $\mathbf{1.577 \pm 0.001}$ |

### 3.3 DO BENEFITS OF NON-NORMAL DYNAMICS EXTEND TO GATED RNN ARCHITECTURES?

So far, we have only considered vanilla RNNs. An important question is whether the benefits of non-normal dynamics demonstrated above for vanilla RNNs also extend to gated RNN architectures like LSTMs or GRUs (Hochreiter & Schmidhuber, 1997; Cho et al., 2014). Gated RNN architectures have better inductive biases than vanilla RNNs in many practical tasks of interest such as language modeling (e.g. see Table 1 for a comparison of vanilla RNN architectures with an LSTM architecture of similar size in the language modeling benchmarks), thus it would be practically very useful if their performance could be improved through an inductive bias for non-normal dynamics.

To address this question, we treated the input, forget, output, and update gates of the LSTM architecture as analogous to vanilla RNNs and initialized the recurrent and input matrices inside these gates in the way as in the chain or the orthogonal initialization of vanilla RNNs above. We also compared these with a more standard initialization scheme where all the weights were drawn from a uniform distribution $\mathcal{U}(-\sqrt{k}, \sqrt{k})$ where $k$ is the reciprocal of the hidden layer size (labeled *plain* in Table 3). This is the default initializer for the LSTM weight matrices in PyTorch: `https://pytorch.org/docs/stable/nn.html#lstm`. We compared these initializers in the language modeling benchmarks. The chain initializer did not perform better than the orthogonal initializer (Table 3), suggesting that non-normal dynamics in gated RNN architectures may not be as helpful as it is in vanilla RNNs. In hindsight, this is not too surprising, because our initial motivation for introducing non-normal dynamics heavily relied on the vanilla RNN architecture and gated RNNs can be dynamically very different from vanilla RNNs.

When we looked at the trained LSTM weight matrices more closely, we found that, although still non-normal, the recurrent weight matrices inside the input, forget, and output gates (i.e. the sigmoid gates) did not have the same signatures of hidden chain-like feedforward structures observed in vanilla RNNs. Specifically, the weight profiles in the LSTM recurrent weight matrices inside these three gates did not display the asymmetric bump characteristic of a prominent chain-like feedforward structure, but were instead approximately monotonic functions of $i - j$ (Figure 6a-c), suggesting a qualitatively different kind of dynamics where the individual units are more persistent over time. The recurrent weight matrix inside the update gate (the `tanh` gate), on the other hand, did display the signature of a hidden chain-like feedforward structure (Figure 6d). When we incorporated these two structures in different gates of the LSTMs, by using a chain initializer for the update gate and a monotonically increasing recurrent weight profile for the other gates (labeled *mixed* in Table 3), the resulting initializer outperformed the other initializers on character-level PTB and `enwik8` tasks.

## 4 DISCUSSION

Motivated by their optimal memory properties in a simplified linear setting (Ganguli et al., 2008), in this paper, we investigated the potential benefits of certain highly non-normal chain-like RNN architectures in capturing long-term dependencies in sequential tasks. Our results demonstrate an advantage for such non-normal architectures as initializers for vanilla RNNs, compared to the commonly used orthogonal initializers. We further found evidence for the induction of such chain-like feedforward structures in trained vanilla RNNs even when these RNNs were initialized with orthogonal recurrent connectivity matrices.

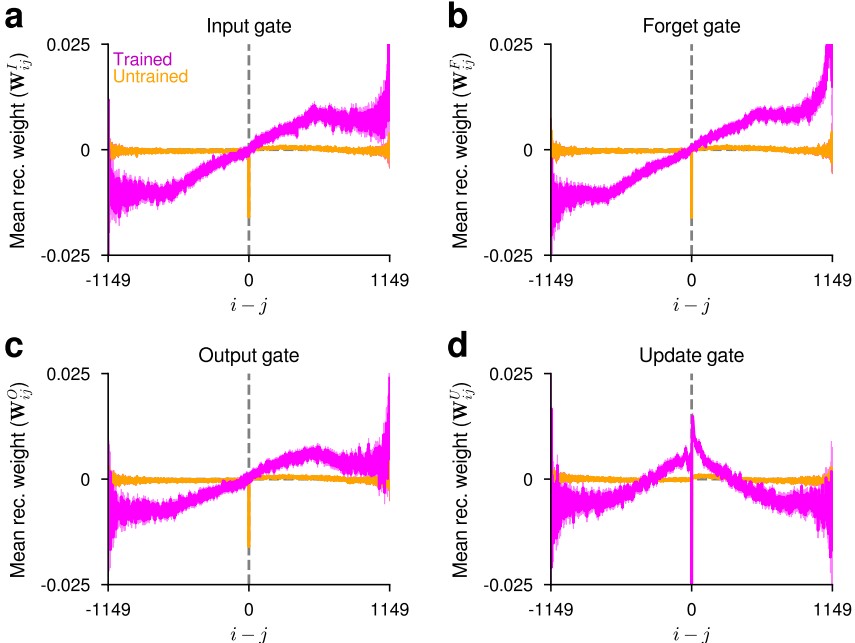

Figure 6: The recurrent weight matrices inside the input, forget, and output LSTM gates do not display the characteristic signature of a prominent chain-like feedforward structure. The weight profiles are instead an approximately monotonic function of $i - j$. The recurrent weight matrix inside the update (`tanh`) gate, however, does display an asymmetric chain-like structure similar to that observed in vanilla RNNs. The examples shown in this figure are from the input (**a**), forget (**b**), output (**c**), and update gates (**d**) of the second layer LSTM in a 3-layer LSTM architecture trained on the word-level PTB task. The weight matrices shown here were initialized with orthogonal initializers. Other layers and models trained on other tasks display qualitatively similar properties.

The benefits of these chain-like non-normal initializers do not directly carry over to more complex, gated RNN architectures such as LSTMs and GRUs. In some important practical problems such as language modeling, the gains from using these kinds of gated architectures seem to far outweigh the gains obtained from the non-normal initializers in vanilla RNNs (see Table 1). However, we also uncovered important regularities in trained LSTM weight matrices, namely that the recurrent weight profiles of the input, forget, and output gates (the sigmoid gates) in trained LSTMs display a monotonically increasing pattern, whereas the recurrent matrix inside the update gate (the `tanh` gate) displays a chain-like feedforward structure similar to that observed in vanilla RNNs (Figure 6). We showed that these regularities can be exploited to improve the training and/or generalization performance of gated RNN architectures by introducing them as useful inductive biases.

A concurrent work to ours also emphasized the importance of non-normal dynamics in RNNs (Kerg et al., 2019). The main difference between Kerg et al. (2019) and our work is that we explicitly introduce sequential motifs in RNNs at initialization as a useful inductive bias for improved long-term memory (motivated by the optimal memory properties of these motifs in simpler cases), whereas their approach does not constrain the shape of the non-normal part of the recurrent connectivity matrix, hence does not utilize *sequential* non-normal dynamics as an inductive bias. In some of their tasks, Kerg et al. (2019) also uncovered a feedforward, chain-like motif in trained vanilla RNNs similar to the one reported in this paper (Figure 5).

There is a close connection between the identity initialization of RNNs (Le et al., 2015) and the widely used identity skip connections (or residual connections) in deep feedforward networks (He et al., 2016). Given the superior performance of chain-like non-normal initializers over the identity initialization demonstrated in the context of vanilla RNNs in this paper, it could be interesting to look for similar chain-like non-normal architectural motifs that could be used in deep feedforward networks in place of the identity skip connections.

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

## A  DETAILS AND EXTENSIONS OF THE LINEAR DECODING EXPERIMENTS

This appendix contains the details of the linear decoding experiments in section 2.2 and reports the results of additional linear decoding experiments. The experiments in section 2.2 compare the signal propagation properties of vanilla RNNs with either random orthogonal or chain connectivity matrices. In both cases, the overall scale of the recurrent connectivity matrices is set to $1.01$. The input weight vector is $\mathbf{v} = [1, 0, 0, \ldots, 0]^\top$ for the chain model and $\mathbf{v}_i \sim \mathcal{N}(0, 1/\sqrt{n})$ for the random orthogonal model (thus the overall scales of both the feedforward and the recurrent inputs are identical in the two models). The RNNs themselves are not trained in these experiments. At each time point, an *i.i.d.* random scalar signal $s_t \sim \mathcal{N}(0, 1)$ is fed into the network as input (Equation 5). We simulate 250 trials for each model and ask how well we can linearly decode the signal at the first time step, $s_1$, from the recurrent activities at time step 100, $\mathbf{h}_{100}$. We do this by linearly regressing $s_1$ on $\mathbf{h}_{100}$ (using the 250 simulated samples) and report the $R^2$ value for the linear regression in Figure 2.

In simulations with noise (Figure 2b), an additional *i.i.d.* random noise term, $z_{it} \sim \mathcal{N}(0, \sigma)$, is added to each recurrent neuron $i$ at each time step $t$. The standard deviation of the noise, $\sigma$, is set to $0.1$ in the experiments shown in Figure 2b. To show that the results are not sensitive to the noise scale, we ran additional experiments with lower ($\sigma = 0.01$) and higher ($\sigma = 1$) levels of noise (Figure 7). In both cases, the chain network still outperforms the orthogonal network. Note that these "linear + noise" experiments satisfy the conditions of the analytical theory in Ganguli et al. (2008), so these results are as expected from the theory.

As mentioned in the main text, the "non-linear + no noise" experiments reported in Figure 2c used the `elu` non-linearity. To show that the results are not sensitive to the choice of the non-linearity, we also ran additional experiments with `tanh` and `relu` non-linearities (Figure 8). As with the `elu` non-linearity, the chain network outperforms the orthogonal network with the `tanh` and `relu` non-linearities as well, suggesting that the results are not sensitive to the choice of the non-linearity.

## B  THE EFFECT OF THE FEEDBACK STRENGTH PARAMETER ($\beta$) IN THE CHAIN WITH FEEDBACK MODEL

In this appendix, we consider the effect of the feedback strength parameter, $\beta$, for the chain with feedback model in the context of the experiments reported in section 3.1.1. We focus on the psMNIST task specifically, because this is the only task where the feedback chain model converges to a low loss solution for a sufficiently large number of hyper-parameter configurations. For the addition and copy tasks, there are not enough successful hyper-parameter configurations to draw reliable inferences about the effect of $\beta$ (see Figure 3d-f). Figure 9 shows the validation loss at the end of training as a function of $\beta$ in the psMNIST task. In this figure, we considered all networks that achieved a validation loss lower than the random baseline model (i.e. $< \log(10) \approx 2.3$) at the end of training (an overwhelming majority of the networks satisfied this criterion). Figure 9 shows that the final validation loss is a monotonically increasing function of $\beta$ in this task, suggesting that large feedback strengths are harmful for the model performance.

## C  COMPARISON WITH PREVIOUS MODELS

In this appendix, we compare our results with those obtained by previous models, focusing specifically on the experiments in section 3.1.1 (because the tasks in this section are commonly used as RNN benchmarks).

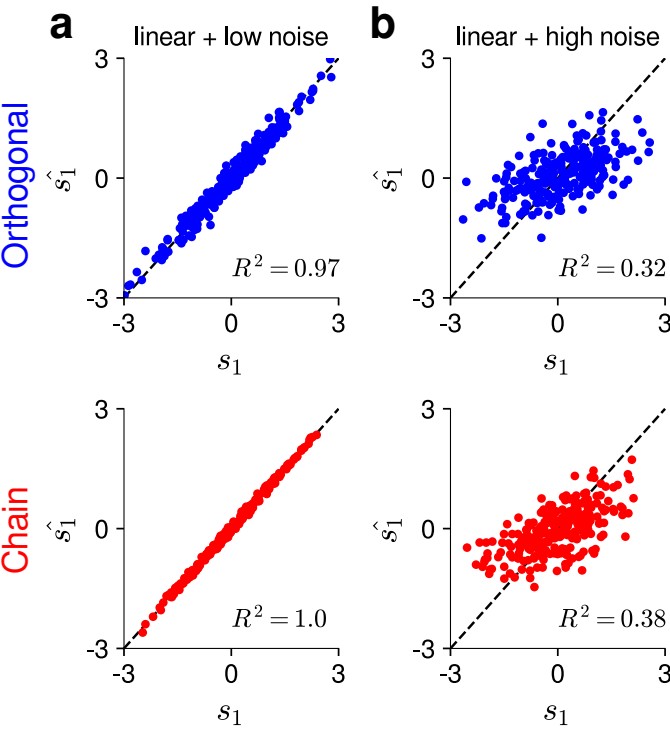

Figure 7: Additional linear decoding experiments: **a** linear networks with low noise ($\sigma = 0.01$) and **b** linear networks with high noise ($\sigma = 1$). In both cases, the chain network outperforms the orthogonal network suggesting that the results are not sensitive to the noise scale.

**uRNN:** We first note that our copy and addition tasks use the largest sequence lengths considered in Arjovsky et al. (2016) for the same tasks ($T = 500$ for the copy task and $T = 750$ for the addition task). Hence our results are directly comparable to those reported in Arjovsky et al. (2016) (the random baselines shown by the dashed lines in Figure 3a-b are identical to those in Arjovsky et al. (2016) for the same conditions). The unitary evolution RNN (uRNN) model proposed in Arjovsky et al. (2016) comfortably learns the copy-500 task (with 128 recurrent units), quickly reaching a near-zero loss (see their Figure 1, bottom right); however, it struggles with the addition task, barely reaching the half-baseline criterion even with 512 recurrent units (see their Figure 2, bottom right). This difference in the behavior of the uRNN model in the copy and addition tasks is predicted by Henaff et al. (2016), where it is shown that random orthogonal and near-identity recurrent connectivity matrices have much better inductive biases in the copy and addition tasks, respectively. Because of its parametrization, uRNN behaves more similarly to a random orthogonal RNN than a near-identity RNN.

In contrast, our non-normal RNNs, especially the chain model, comfortably clear the half-baseline criterion both in copy-500 and addition-750 tasks (with 100 recurrent units), quickly achieving very small loss values in both tasks with the optimal hyper-parameter configurations (Figure 3a-b). Note that this is despite the fact that our models use fewer recurrent units than the uRNN model in Arjovsky et al. (2016) (100 vs. 128 or 512 recurrent units).

**nnRNN:** Kerg et al. (2019) report results for the copy ($T = 200$) and psMNIST tasks only. They have not reported training success for longer variants of the copy task (specifically for $T = 500$). Kerg et al. (2019) also have not reported successful training in the addition task, whereas our non-normal RNNs showed training success both in copy-500 and addition-750 tasks (Figure 3a-b).

We conclude that our non-normal initializers for vanilla RNNs perform comparably to, or better than, the uRNN and nnRNN models in standard long-term memory benchmarks. One of the biggest strengths of our proposal compared to these previous models is its much greater simplicity. Both uRNN and nnRNN require a complete re-parametrization of the vanilla RNN model (nnRNN even

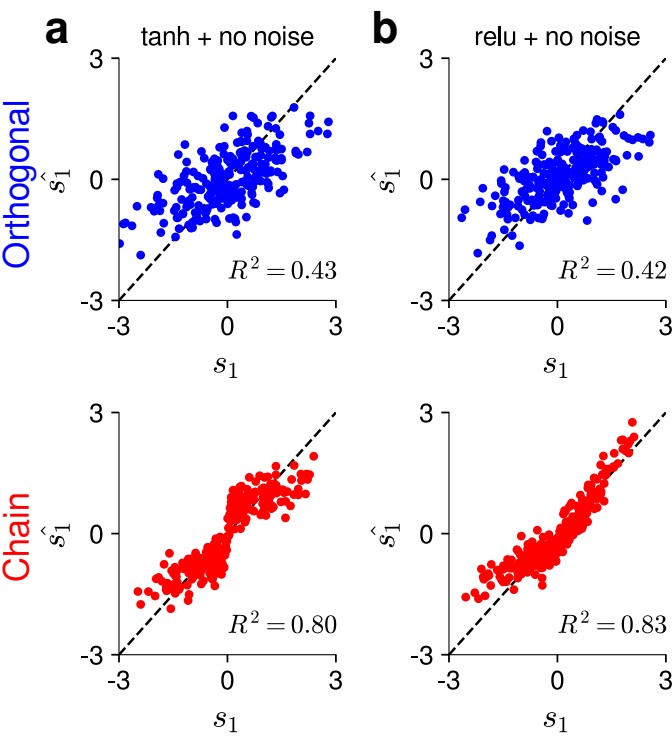

Figure 8: Additional linear decoding experiments: **a** `tanh` networks with no noise and **b** `relu` networks with no noise. In both cases, the chain network outperforms the orthogonal network suggesting that the results are not sensitive to the choice of the non-linearity.

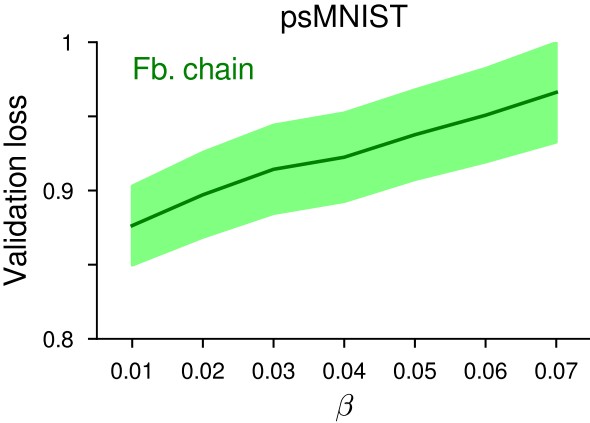

Figure 9: Validation loss at the end of training as a function of the feedback parameter $\beta$ in the psMNIST task. All networks with a better-than-random loss at the end of training are included in this figure. The solid line shows the mean and the shaded region represents the standard errors.

requires a novel optimization method). Our method, on the other hand, proposes much simpler, easy-to-implement, plug-and-play type sequential initializers that keep the standard parametrization of RNNs intact.

**critical RNN:** Chen et al. (2018) note that the conditions for dynamical isometry in vanilla RNNs are identical to those in fully-connected feed-forward networks studied in Pennington et al. (2017). Pennington et al. (2017), in turn, note that dynamical isometry is not achievable exactly in networks

with `relu` activation, but it is achievable in networks with `tanh` activation, where it essentially boils down to initializing the weights to small values. Pennington et al. (2017) give a specific example of a dynamically isometric `tanh` network (with $n = 400$, $\sigma_w = 1.05$, and $\sigma_b = 2.01 \times 10^{-5}$). We set up a similar `tanh` RNN model, but were not able to train it successfully in the copy or addition tasks. Again, as with the nnRNN results, this shows the challenging nature of these two tasks and suggests that dynamical isometry may not be enough for successful training in these tasks. A possible reason for this is that although critical initialization takes the non-linearity into account, it still does not take the noise into account (i.e. it is not guaranteed to maximize the SNR).

**LSTM, `tanh` RNN:** Consistent with the results in Arjovsky et al. (2016), we were not able to successfully train LSTMs or vanilla RNNs with `tanh` non-linearity in the challenging copy-500 and addition-750 tasks. Therefore, these models were not included as baselines in section 3.1.1.

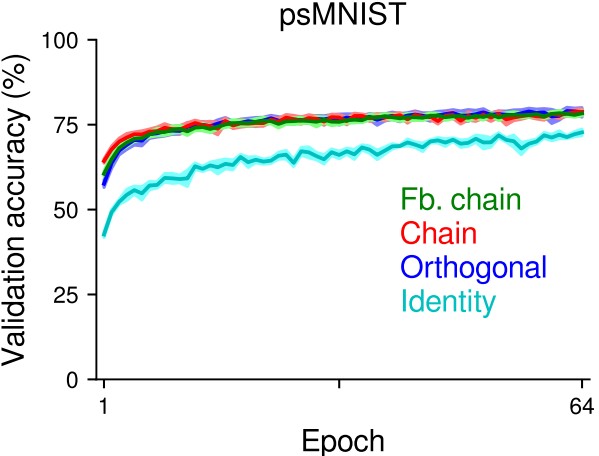

Figure 10: Validation accuracy in the psMNIST task. The corresponding validation losses are shown in Figure 3c in the main text. Note that we used RNNs with $n = 25$ recurrent units in these simulations, so these numbers are not directly comparable to those reported in some previous works (e.g. Arjovsky et al. (2016); Kerg et al. (2019)).

