# OpenReview forum: "Improved memory in recurrent neural networks with sequential non-normal dynamics"
_ICLR.cc/2020/Conference — Accept (Poster)_

### Official Review · AnonReviewer2 · 2019-10-23
**Official Blind Review #2**

**Rating:** 6

**Review:**

Contributions:
 This paper proposes to explore nonnormal matrix initialization in RNNs. Authors demonstrate on various tasks (Copy/Addition, Permuted-SMNIST, PTB, enwik8) that chain-like nonnormal matrix initializations can outperform orthogonal or identity initialization in vanilla RNNs. However, nonnormal RNNs underperform their gated counterpart such as LSTM. Authors also show results where they use their initialization scheme in update gate of a LSTM.

Comments:
The paper is well written and pleasant to read. The paper structure could be a bit improved. For instance, section 2 is named “Results” while 2.1 which take significant part of the section is about some prior results in (Ganguli et al. 2018). It would be better to have it under an explicit prior work section.

The description of the experiments reported in Figure 2. is a bit vague: what is the training/evaluation data?, do you train all the model parameters or only the linear layer?, what is the type of noise used? It is unclear to me how robust is the observation made in Figure-2. Do you see similar behavior with different noise-scale and other non-linearity such as tanh?

The experimental section provides convincing data showing that non-normal initialization schemes outperform orthogonal and identity initialization in vanilla RNN. However, it would be nice to add some comparisons with prior works. It is unclear how the current method compare with nn-RNN of (Kerg et al. 2019) and the unitary-RNNs.

 Why the score reported for the 3-LSTM in Table 3.  is underperforming 3-layer LSTM baseline used in (Merity et al., 2018), reported in Table 1.?  In addition, did you try saturating non-linearities for the RNN experiments?

Overall, I think the method is promising, but comparison with prior work is missing. I would encourage the authors to compare their approach with unitary-RNN, and nn-RNN to better demonstrate the significance of their works.

Additional remarks:
-	SNR could be defined more precisely in the introduction. In particular, the introduction states that the stochasticity of SGD is a source of noise which is true. But the model presented in section 2 seems to focus mostly on input noise?


**Experience Assessment:**

I have published one or two papers in this area.

**Review Assessment: Checking Correctness Of Derivations And Theory:**

I assessed the sensibility of the derivations and theory.

**Review Assessment: Checking Correctness Of Experiments:**

I carefully checked the experiments.

**Review Assessment: Thoroughness In Paper Reading:**

I read the paper thoroughly.

---

> ### Author Response · Authors · 2019-11-13
> **response to reviewer 2**
>
> Thank you very much for your review. We particularly appreciate your encouraging comments, such as: “the paper is well written and pleasant to read”, “the method is promising”, and “the experimental section provides convincing data showing that non-normal initialization schemes outperform orthogonal and identity initialization in vanilla RNN”. Please find below our responses to the specific questions and concerns raised in your review.
>
> 1) “The paper structure could be a bit improved. For instance, section 2 is named “Results” while 2.1 which take significant part of the section is about some prior results in (Ganguli et al. 2018). It would be better to have it under an explicit prior work section.”
>
> Thank you for this suggestion. The paper has been updated along these lines. Specifically, we renamed section 2 as “Background” and section 3 as “Results”.
>
> 2) “The description of the experiments reported in Figure 2. is a bit vague: what is the training/evaluation data? do you train all the model parameters or only the linear layer?, what is the type of noise used? It is unclear to me how robust is the observation made in Figure-2. Do you see similar behavior with different noise-scale and other non-linearity such as tanh?”
>
> We have added an appendix to the paper where we describe these experiments in much greater detail (Appendix A). To answer your questions briefly, the RNNs themselves are not trained in these experiments and the noise is iid Gaussian. We have also run additional experiments to show that the results are robust to the noise scale and different choices of the nonlinearity (tanh and relu). Please see Appendix A for further details.
>
> 3) “The experimental section provides convincing data showing that non-normal initialization schemes outperform orthogonal and identity initialization in vanilla RNN. However, it would be nice to add some comparisons with prior works. It is unclear how the current method compare with nn-RNN of (Kerg et al. 2019) and the unitary-RNNs.”
>
> We have added another appendix (Appendix C) where we compare our method with previous methods in greater detail. Briefly, we conclude that our non-normal initializers perform comparably to, or better than, uRNN and nnRNN in standard long-term memory benchmarks. The biggest advantage of our proposal over these previous methods is its much greater simplicity. Both uRNN and nnRNN require a complete re-parametrization of the vanilla RNN model (nnRNN even requires a novel optimization method). Our method, on the other hand, proposes much simpler, easy-to-implement, plug-and-play type sequential initializers that keep the standard parametrization of RNNs intact.
>
> 4) “Why the score reported for the 3-LSTM in Table 3 is underperforming 3-layer LSTM baseline used in (Merity et al., 2018), reported in Table 1?”
>
> Our training setup was optimized for training vanilla RNNs (instead of LSTMs), hence it differs slightly from Merity et al. (2018) as described in points 2 and 3 in section 3.1.2 of the updated paper. The differences between Tables 1 and 3 are thus attributable to these differences in the training setups.
>
> 5) “In addition, did you try saturating non-linearities for the RNN experiments?”
>
> Yes, we did try RNNs with tanh non-linearity in the experiments described in section 3.1.1 of the updated paper. These models did not appear to perform better than chance in the copy-500 and addition-750 tasks (with either standard or non-normal initializers), suggesting that a non-linearity that is linear over a sufficiently wide input range, such as elu or relu, might be necessary for the success of the non-normal initializers.
>
> 6) “SNR could be defined more precisely in the introduction. In particular, the introduction states that the stochasticity of SGD is a source of noise which is true. But the model presented in section 2 seems to focus mostly on input noise?”
>
> This is correct. The analytical theory developed in Ganguli et al. (2008) assumes linear networks and iid Gaussian input noise. The motivation behind our experiments is thus to test how well this theory generalizes to more realistic problems by incorporating a nonlinearity in the networks and considering practically more relevant sources of noise such as the noise induced by SGD during training (which, as you correctly point out, is not iid Gaussian in general).

---

> > ### Comment · AnonReviewer2 · 2019-11-15
> > **Thanks for the rebuttal**
> >
> > Thanks for the nice rebuttal and the additional experiments!
> >
> > The rebuttal addresses most of my initial concerns, I updated my rating to reflect this.

---

### Official Review · AnonReviewer3 · 2019-10-23
**Official Blind Review #3**

**Rating:** 8

**Review:**

The focus of this paper is on exploring non-normal initializations for training vanilla RNN for sequential tasks. They show on 3 different tasks, and a real-world LM task that  non-normal initializations of vanilla RNNs outperform their orthogonal counter-parts when particular forms of initialization are considered.


Although the results for sequence task do not outperform the gated counterparts, the authors present an interesting exploration of initializing non-normal RNNs that outperform the orthogonal counterparts. It is good to see this line of work being explored as an alternative to exploring more complex architectures with many more parameters than necessary for the task.

Strengths:
    1. The paper explores non-normal RNNs and demonstrates on  3 synthetic tasks - copy, addition and pMNIST - how with careful initialization the proposed approach outperforms their orthogonal initialization counterpart. This line of experimentation is interesting as it potentially opens the door for more expressive modeling for sequential tasks by expanding the solution space of the weight matrices being learnt i.e orthogonal matrices are a special case.
    2. The authors do a great job in motivating the paper, and the explanation is clear and easily understandable. The toy simulations in Section2.2 really helps drive the reasoning behind why chain initialization improves over orthogonal initialization.
    3. Based on the insight from trained RNNs where the trained  weights exhibit a chain like structure, the authors attempt to modify the LSTM gate initializations well. However, they do not see any specific gain by doing so, and moreover they show analysis that demonstrate that the LSTM gates do not learn these chain like structures. However, they do have insight into the regularities of these learnt weights which could potentially open the door for more interesting initialization methods for training such gated architectures.


Issues to be addressed in the paper:
1. The plots are quite small and hard to follow. Can the authors enlarge these so they span the entire page? Also, for pMNIST it would be good to provide accuracy scores as well as a function of the training epochs.Finally, it would be good to include a comparison against LSTMs (and even Transformer networks) so it is easier for the reader to see where these approaches stack against architecture changes.
2. The authors are missing a reference to this  work - http://proceedings.mlr.press/v48/henaff16.pdf  - which provides empirical analysis for the 3 synthetic tasks to test the ability of vanilla RNNs for solving long span sequential tasks.
3. What about stability of these non-normal RNNs? For example, if we perturb the inputs to the training for the LM task how much variance do we see in the performance of these models?


**Experience Assessment:**

I do not know much about this area.

**Review Assessment: Checking Correctness Of Derivations And Theory:**

N/A

**Review Assessment: Checking Correctness Of Experiments:**

I carefully checked the experiments.

**Review Assessment: Thoroughness In Paper Reading:**

I read the paper thoroughly.

---

> ### Author Response · Authors · 2019-11-13
> **response to reviewer 3**
>
> Thank you very much for your review and for your encouraging comments. Please find below our responses to the specific issues you raise in your review.
>
> 1) “The plots are quite small and hard to follow. Can the authors enlarge these so they span the entire page?”
>
> In the updated paper, we tried our best to enlarge all figures as much as possible given the space constraints.
>
> 2) “Also, for pMNIST it would be good to provide accuracy scores as well as a function of the training epochs.”
>
> We have added this figure to the appendix in the updated paper (see Figure 10), as you requested.
>
> 3) “Finally, it would be good to include a comparison against LSTMs (and even Transformer networks) so it is easier for the reader to see where these approaches stack against architecture changes.”
>
> Consistent with previous reports (e.g. Arjovsky et al., 2016), we observed that LSTMs do not perform much better than chance in the copy-500 and addition-750 tasks, hence we did not include them as baselines in the experiments reported in section 3.1.1.
>
> 4) “The authors are missing a reference to this work: http://proceedings.mlr.press/v48/henaff16.pdf which provides empirical analysis for the 3 synthetic tasks to test the ability of vanilla RNNs for solving long span sequential tasks.”
>
> In the updated paper, we have added a reference to this work, citing it in connection with the differing behavior of the uRNN model of Arjovsky et al. (2016) in the copy-500 and addition-750 tasks (see Appendix C).
>
> 5) “What about stability of these non-normal RNNs? For example, if we perturb the inputs to the training for the LM task how much variance do we see in the performance of these models?”
>
> This is a good idea, but unfortunately, we did not have enough time to perform these perturbation experiments in the LM tasks during the rebuttal period (as these experiments take quite a bit of time to perform thoroughly). However, we will be happy to include these results in the final camera-ready version. Relatedly, please do note that robustness to noise, or maximizing the signal-to-noise ratio (SNR), is the explicit motivation behind the introduction of these non-normal, sequential motifs to RNNs, so we expect our non-normal RNNs to be more robust against noise than alternative models. Figure 2 in the paper provides a simple demonstration of this: the estimates from the chain RNN have less variance than the estimates from the orthogonal RNN in these experiments.

---

### Official Review · AnonReviewer1 · 2019-10-26
**Official Blind Review #1**

**Rating:** 3

**Review:**

Motivated by the sub-optimality of using orthogonal recurrent  matrix in RNNs with nonlinearity and noise, the authors look into non-normal alternatives, in particular matrices with chain-like structure in preserving memory in RNNs. The authors compare normal and non-normal RNNs on several sequential benchmark datasets, and show that non-normal RNNs perform better than their normal counterpart.

The paper is easy to follow. The novelty of the work is limited though. The chain structure was introduced in Ganguli et al. (2008). The work studies the benefit of initializing recurrent weights in nonlinear RNNs with these chain-like structures.

Chen et. al. (2018) already pointed out the limitation of orthogonal initialization alone for nonlinear RNNs, and proposed closed-form initialization for RNNs with different activation functions. It would be worthwhile to include a comparison to that method.

In experiment section 2.3.1, it would be helpful to include comparison of performance of chain with feedback using different beta values to confirm the intuition that stronger feedback strength would negatively impact the memory.

Results in section 2.3.2 Table 1 are not exactly align with the story. Do the authors have any intuition on why the chain with feedback perform better than the chain variant.


**Experience Assessment:**

I have published one or two papers in this area.

**Review Assessment: Checking Correctness Of Derivations And Theory:**

I assessed the sensibility of the derivations and theory.

**Review Assessment: Checking Correctness Of Experiments:**

I assessed the sensibility of the experiments.

**Review Assessment: Thoroughness In Paper Reading:**

I read the paper at least twice and used my best judgement in assessing the paper.

---

> ### Author Response · Authors · 2019-11-13
> **response to reviewer 1**
>
> Thank you very much for your review. Please find below our detailed responses to the questions and concerns raised in your review.
>
> 1) “The paper is easy to follow. The novelty of the work is limited though. The chain structure was introduced in Ganguli et al. (2008). The work studies the benefit of initializing recurrent weights in nonlinear RNNs with these chain-like structures.”
>
> We note that our contributions in the paper are not limited to introducing chain-like motifs to nonlinear RNNs as a useful inductive bias. We also uncover similar motifs in trained vanilla RNNs initialized with orthogonal recurrent connectivity matrices (section 3.2 in the updated paper) and discover two qualitatively different types of structure inside the sigmoid vs. tanh gates of trained LSTMs (section 3.3 in the updated paper).
>
> 2) “Chen et. al. (2018) already pointed out the limitation of orthogonal initialization alone for nonlinear RNNs, and proposed closed-form initialization for RNNs with different activation functions. It would be worthwhile to include a comparison to that method.”
>
> Thank you for pointing out this reference. In the updated paper, we have acknowledged this paper (as well as the related paper by Pennington et al. (2017)) in the Introduction, and added a discussion of our attempts at training critically initialized RNNs in Appendix C.
>
> 3) “In experiment section 2.3.1, it would be helpful to include comparison of performance of chain with feedback using different beta values to confirm the intuition that stronger feedback strength would negatively impact the memory.”
>
> As suggested by the reviewer, we have added an analysis of the effect of the feedback strength, beta, to the paper (please see Appendix B and the corresponding Figure 9). This analysis confirms the reviewer’s intuition that “stronger feedback strength would negatively impact the memory”. However, please note that we were able to carry out this analysis only for the psMNIST task, because for the other tasks, there were not enough “successful” hyper-parameter configurations for the feedback chain model to draw reliable conclusions about the effect of beta (see Figure 3d-f).
>
> 4) “Results in section 2.3.2 Table 1 are not exactly align with the story. Do the authors have any intuition on why the chain with feedback perform better than the chain variant.”
>
> This is an interesting question. We first note that maintaining a long-term memory is useful in language modeling tasks, but it is not sufficient. Another important component of these tasks is modeling both long-term and short-term contextual dependencies (e.g. what words are more likely given a particular context that occurred some time steps ago). In language modeling, short-term dependencies are stronger than longer-term dependencies and we conjecture that being able to model these relatively short-term dependencies well enough is more important for the model performance than maintaining a very long-term memory (we believe this may also be the reason why gated RNN architectures generally outperform vanilla RNNs in these tasks). Very sparse recurrent connectivity matrices like the chain model (Fig. 1a “Chain”) may have reduced initial expressivity compared to denser variants such as the  chain with feedback model (Fig. 1a “Chain with feedback”). This may, in turn, cause the chain variant to perform worse in capturing a large number of short-term contextual dependencies. Designing more expressive, less sparse non-normal connectivity matrices that nevertheless retain the optimal memory properties of the sequential connectivity matrices is an interesting future direction.

---

### Author Response · Authors · 2019-11-13
**revision uploaded**

We thank all three reviewers for their thoughtful comments. We have uploaded a revised version of our paper that addresses the issues raised by the reviewers. Our detailed responses to the specific questions and concerns raised by the reviewers can be found below.

---

### Decision · Program_Chairs · 2019-12-19

**Decision:**

Accept (Poster)

**Comment:**

This paper proposes to explore nonnormal matrix initialization in RNNs.  Two reviewers recommended acceptance and one recommended rejection.  The reviewers recommending acceptance highlighted the utility of the approach, its potential to inspire future work, and the clarity and quality of writing and accompanying experiments.  One reviewer recommending weak acceptance expressed appreciation of the quality of the rebuttal and that their concerns were largely addressed.  The reviewer recommending rejection was primarily concerned with the novelty of the method.  Their review suggested the inclusion of an additional citation, which was included in a revised version for the rebuttal but not with a direct comparison of results.  On the balance, the paper has a relatively high degree of support from the reviewers, and presents an interesting and potentially useful initialization in a clear and well-motivated way.